# The Usefulness of the Athens Insomnia Scale for Evaluating Sleep Disturbance in Patients with Chronic Liver Disease Comparing with Pittsburgh Sleep Quality Index and Epworth Sleepiness Scale

**DOI:** 10.3390/medicina58060741

**Published:** 2022-05-30

**Authors:** Hideto Kawaratani, Hisamitsu Miyaaki, Atsushi Hiraoka, Kazuhiko Nakao, Yoichi Hiasa, Hitoshi Yoshiji, Kiwamu Okita, Kazuhiko Koike

**Affiliations:** 1Department of Gastroenterology, Nara Medical University, Nara 634-8522, Japan; yoshijih@naramed-u.ac.jp; 2Department of Gastroenterology and Hepatology, Nagasaki University Graduate School of Biomedical Sciences, Nagasaki 852-8523, Japan; miyaaki-hi@nagasaki-u.ac.jp (H.M.); kazuhiko@nagasaki-u.ac.jp (K.N.); 3Gastroenterology Center, Ehime Prefectural Central Hospital, Matsuyama 790-0024, Japan; hirage@gmail.com; 4Department of Gastroenterology and Metabology, Ehime University Graduate School of Medicine, Toon 791-0295, Japan; hiasa@m.ehime-u.ac.jp; 5Shunan Memorial Hospital, Yamaguchi 744-0033, Japan; icb68895@nifty.com; 6Kanto Central Hospital, Tokyo 158-8531, Japan; kkoike.tky@gmail.com

**Keywords:** Athens insomnia scale, hepatic encephalopathy, sleep disturbance

## Abstract

*Background and Objectives*: Sleep disturbance due to muscle cramps or hepatic encephalopathy in patients with chronic liver disease (CLD) can lead to a reduced quality of life. The Pittsburgh sleep quality index (PSQI) is commonly used for evaluating sleep disturbance; however, this questionnaire is time-consuming owing to the large number of questions. As the usefulness of the Athens insomnia scale (AIS) in patients with CLD is not sufficiently known, the present study aimed to determine whether the AIS and Epworth sleepiness scale (ESS) could be used as simple alternative questionnaires for evaluating sleep disturbances in patients with CLD. *Materials and Methods*: A total of 117 patients with CLD were retrospectively evaluated. Patients with overt hepatic encephalopathy were excluded. All patients were examined using the AIS, PSQI, and ESS, and their responses to these questionnaires were statistically analyzed. *Results*: The number of patients diagnosed with sleep disturbance using the AIS, PSQI, and ESS were 39 (33.3%), 37 (31.6%), and 9 (7.7%), respectively. There was no correlation between PSQI and ESS scores (*r* = 0.011, *p* = 0.910); in contrast, the AIS scores showed a significant correlation with the PSQI scores (*r* = 0.689, *p* < 0.001). When the PSQI was considered as the standard for evaluating sleep disturbance, the sensitivity, specificity, positive predictive value (PPV), and negative predictive value (NPV) of the AIS were 76.9%, 91.0%, 81.1%, and 88.8%, respectively. In the sleep medication group, the sensitivity, specificity, PPV, and NPV of the AIS were 100%, 70%, 78.6%, and 100%, respectively. *Conclusions*: This is the first report to indicate that the AIS is an alternative questionnaire to the PSQI and that it can be a useful tool for detecting cirrhosis-related complications in patients with CLD.

## 1. Introduction

Chronic liver disease (CLD) is an important public health problem that can lead to liver cirrhosis and hepatocellular carcinoma. Liver cirrhosis is the 11th leading cause of death globally, while hepatocellular carcinoma is the 16th [1]. Viral infection, such as hepatitis B or C, alcohol abuse, non-alcoholic steatohepatitis, and autoimmunity induce CLD. Patients with CLD experience reduced quality of life (QOL) owing to ascites, hepatic encephalopathy (HE), muscle cramps, and fatigue [2,3]. Sleep disturbance also reduces QOL [4,5], and muscle cramps and covert HE are known causes of sleep disturbances in patients with CLD [4,6,7]. The prevalence of sleep disturbance is larger in compensated cirrhotic patients compared to healthy controls [8]

Sleep disturbance is often treated with sleep medications; however, these medications inhibit the central nervous system and can induce HE, which alters consciousness, mood, or personality in patients with advanced CLD [7,9]. Therefore, treatment for sleep disturbance in patients with CLD should be performed in accordance with the underlying cause, and prescribing sleep medications should be avoided owing to the risk of HE.

The Pittsburgh sleep quality index (PSQI) is commonly used to assess sleep disturbance [10], and its utility for evaluating patients with CLD is well-established [11]. However, this questionnaire is time-consuming, owing to the large number of questions (19 items), and its contents are subjective. The contents of the PSQI are presented in Figure 1. On the other hand, the Athens insomnia scale (AIS) is a simple questionnaire that has been validated in the general population, psychiatric patients, and patients with insomnia [12,13,14]. The contents of the AIS are presented in Figure 2. As the AIS contains only eight items and its contents are objective, it can be used as a convenient and useful diagnostic tool based on the International Classification of Diseases-10th revision [15]. The Epworth sleepiness scale (ESS) is used to assess daytime sleepiness [16] and has been validated in a number of languages and clinical settings [17]. The ESS also contains eight items; the contents of the ESS are shown in Figure 3. Nonetheless, the utility of the AIS and ESS for evaluating sleep disturbance in patients with CLD has not yet been validated [18]. The present study aimed to clarify the utility of the AIS and ESS for evaluating sleep disturbance in patients with CLD.

## 2. Materials and Methods

### 2.1. Study Design and Patients

The present study was a retrospective, multicenter, cross-sectional study of 117 patients with CLD who were evaluated using the AIS, PSQI, and ESS. Outpatients or hospitalized patients aged > 20 years who had chronic hepatitis or liver cirrhosis were included in this study. Patients with overt HE at the time of evaluation or neurological/psychiatric diseases were excluded from the analysis. Patients were enrolled from 1 January 2021 to 30 June 2021. All patients were Japanese. The Nara Medical University Hospital, the Ehime Prefectural Central Hospital, and the Nagasaki University Hospital collaborated in this study. The study was approved by the ethics committee of each institution (approval no. 20091406-3) in accordance with the ethical guidelines of the 1975 Declaration of Helsinki. Informed consent was obtained from all patients.

### 2.2. Sleep Quality Questionnaires

The PSQI is the gold standard for assessing sleep quality over a period of 1 month. This questionnaire includes 19 separate items and the following seven components: sleep quality, sleep latency, sleep duration, sleep efficiency, sleep disturbances, sleep medication, and daytime dysfunction. Each component is weighted equally and scored on a scale of 0–3 for a total score of 0–21. The contents of the PSQI are shown in Figure 1. The PSQI takes approximately 10 min to answer and 5 min to score. Higher scores represent subjective sleep quality disorder. Patients were classified into normal sleeping habit (PSQI score: 0–5 points), mild sleep disorder (6–8 points), moderate sleep disorder (9–11 points), and severe sleep disorder (12 points or greater) [19]. PSQI has been validated in various languages and clinical settings [20]. In this study, sleep disturbance was defined as a total score of ≥9 is indicative of sleep disturbance. The AIS is used to evaluate sleep disturbance and consists of eight questions rated on a scale of 0 (no problem at all) to 3 (extremely problematic). The contents of the AIS are shown in Figure 2. The total score ranges from 0–24, with a total score of 0–3 as normal, 4–6 indicative of subclinical insomnia, and ≥6 indicative of clinical insomnia [14]. AIS has been validated in the general population, psychiatric patients, and patients with insomnia [12,13,14]. In this study, sleep disturbance was defined as a total score of ≥6. The ESS comprises eight items related to falling asleep or sleepiness while engaging in seven different daily living activities and is used to assess excessive daytime sleepiness. The contents of the ESS are shown in Figure 3. The ESS score ranges from 0 to 24, and higher scores are indicative of more time spent in a sleep state during the day [16] and has been validated in a number of languages and clinical settings [17]. Sleep disturbance is defined as a total score of ≥11.

### 2.3. Statistical Analysis

Statistical analyses were performed using Student’s t-test or the Mann–Whitney U test. Spearman’s test and Cohen’s kappa coefficient of agreement were used to assess relationships between the tests. Statistical significance was set at *p* < 0.05. Statistical analyses were performed using Easy R (EZR) version 1.54 (Saitama Medical Center, Jichi Medical University, Saitama, Japan) [21].

## 3. Results

### 3.1. Prevalence of Sleep Disturbance

A total of 117 patients (78 men, 39 women) were evaluated in this study. The mean patient age was 69.1 years. Of these patients, 104 had cirrhosis, whereas 73 had hepatocellular carcinoma. Sleep disturbance was identified in 39 (33.3%), 37 (31.6%), and 9 (7.7%) patients based on AIS, PSQI, and ESS scores, respectively (Figure 4).

### 3.2. Baseline Characteristics of PSQI and AIS Scores

The baseline characteristics of PSQI scores are summarized in Table 1. Patients with sleep disturbance and those with normal sleep significantly differed with respect to NH_3_ (72.3 ± 51.9 vs. 51.5 ± 30.2 μg/dL, *p* < 0.05), administration of sleep medication (37.8% vs. 8.8%, *p* < 0.001), branched chain amino acid (BCAA) (51.4% vs. 30.0%, *p* < 0.05), and carnitine (8.1% vs. 0%, *p* < 0.05). Table 2 presents the baseline characteristics of the AIS scores. There was a significant difference in age (66.4 ± 9.1 vs. 70.5 ± 12.3 years, *p* < 0.05) and administration of Shakuyakukanzoto (Kampo medicine for muscle cramps) (23.1% vs. 3.8%, *p* < 0.01) between patients with sleep disturbance and those with normal sleep. Table 3 presents the baseline characteristics of the combination of PSQI and AIS scores. There was a significant difference in age (66.1 ± 8.2 vs. 70.5 ± 12.2, *p* < 0.05), presence of liver cirrhosis (100% vs. 84.5%, *p* < 0.05), administration of sleep medication (36.7% vs. 9.9%, *p* < 0.01), administration of Shakuyakukanzoto (16.7% vs. 4.2%, *p* < 0.05), and BCAA (53.3% vs. 29.6%, *p* < 0.05) between patients diagnosed with sleep disturbance according to the AIS and PSQI scores and those with normal sleep.

### 3.3. Relationships among Questionnaires

In the entire study cohort, relationships were observed between AIS and PSQI scores (*r* = 0.689, *p* < 0.001) and between AIS and ESS scores (*r* = 0.204, *p* = 0.027); however, there was no significant relationship between ESS and PSQI scores (*r* = 0.011, *p* = 0.910). In patients who received sleep medication, there was a relationship between AIS and PSQI scores (*r* = 0.742, *p* < 0.001) but not between AIS and ESS scores (*r* = 0.213, *p* = 0.353) and ESS and PSQI scores (*r* = 0.158, *p* = 0.494). In patients who did not receive sleep medication, relationships were noted between AIS and PSQI scores (*r* = 0.660, *p* < 0.001) and between AIS and ESS scores (*r* = 0.221, *p* = 0.030) but not between ESS and PSQI scores (*r* = 0.007, *p* = 0.943).

### 3.4. Comparison between the AIS and PSQI

A total of 80 patients were determined to have normal sleep according to the PSQI scores, and 37 were determined to have sleep disturbance (score of ≥9). Similarly, 78 patients were determined to have normal sleep according to the AIS scores, and 39 were determined to have sleep disturbance (score of ≥6) (Table 4). When the PSQI was considered as the standard for evaluating sleep disturbance, the sensitivity, specificity, positive predictive value (PPV), negative predictive value (NPV), and kappa coefficient were 76.9%, 91.0%, 81.1%, 88.8%, and 0.68, respectively.

We further divided the cohort into the following two groups: those who received sleep medication and those who did not. In the sleep medication group, 7 and 14 patients were determined to have normal sleep and sleep disturbance, respectively, according to the PSQI scores, whereas 10 and 11 patients were determined to have normal sleep and sleep disturbance, respectively, according to the AIS scores. The sensitivity, specificity, PPV, NPV, and kappa coefficient of the AIS were 100.0%, 70.0%, 78.6%, 100.0%, and 0.68, respectively. In the no sleep medication group, 73 and 23 patients were determined to have normal sleep and sleep disturbance, respectively, according to the PSQI scores, whereas 68 and 28 patients were determined to have normal sleep and sleep disturbance, respectively, according to the AIS scores. The sensitivity, specificity, PPV, NPV, and kappa coefficient of the AIS were 67.9%, 94.1%, 82.6%, 87.7%, and 0.65, respectively.

### 3.5. Comparison of the ESS and PSQI or AIS

According to the ESS, 108 patients had normal sleep, and nine patients had sleep disturbance (Table 5). When the PSQI was considered as the standard for evaluating sleep disturbance, the sensitivity, specificity, PPV, NPV, and kappa coefficient of the ESS were 33.3%, 68.5%, 92.5%, 8.1%, and 0.008, respectively. When the AIS was considered as the standard for evaluating sleep disturbance, the sensitivity, specificity, PPV, NPV, and kappa coefficient of the ESS were 66.7%, 68.6%, 15.4%, 96.2%, and 0.14, respectively. We added the Venn diagram for summarize the relationship between AIS, PSQI, and ESS (Figure 5).

## 4. Discussion

This study aimed to clarify the utility of the AIS and ESS for evaluating sleep disturbance in patients with CLD. We found that the AIS could be used as a simple alternative questionnaire to the PSQI. In contrast, the ESS was not correlated with the AIS or PSQI. To our knowledge, this is the first study to compare the utility of the AIS, PSQI, and ESS for evaluating sleep disturbance in patients with CLD. In this study, the prevalence rates of sleep disturbance were 33.3%, 31.6%, and 7.7% according to the AIS, PSQI, and ESS scores, respectively. These findings indicate that the prevalence of daytime sleepiness was low in patients with CLD and that patients without liver cirrhosis experienced less sleep disturbance. The prevalence of sleep disturbance that we obtained based on the AIS and PSQI scores was slightly higher than that in the general Japanese population (21.4%) [22], but it was similar to that previously reported in patients with CLD [4,5,23].

As the PSQI was developed in 1988, several studies have investigated sleep disturbance according to the PSQI in patients with CLD [11,24,25]. However, this questionnaire is time-consuming owing to the large number of questions, and its contents are subjective. Comparatively few studies have investigated sleep disturbance, as evaluated by the AIS and ESS, in patients with CLD. A previous study reported a strong correlation between the PSQI and AIS and a weak correlation between the PSQI and ESS in Korean firefighters [26]. Nevertheless, there are no data pertaining to the relationship between the PSQI and AIS or the PSQI and ESS in patients with CLD. The present study revealed a significant correlation between the PSQI and AIS but no correlation between the PSQI and ESS. Thus, AIS could be used as an alternative to the PSQI in patients with CLD.

Treating liver-related complications such as covert HE and muscle cramps can improve sleep disturbance in patients with CLD [27,28]. Our previous study revealed that muscle cramps occurred at night, interrupted sleep, and reduced sleep quality [6]. Furthermore, sleep disturbance is associated with muscle strength independent of skeletal muscle loss [29]. Evaluation of skeletal muscle strength is also useful for sleep disturbance. Daytime sleepiness is a common finding in covert HE, and the treatment of covert HE improves daytime sleepiness [23,30]. Patients with overt HE-related sleep disturbance were more likely to experience daytime sleepiness and poor sleep quality. In addition, we previously reported that a high percentage (51.8%) of patients with CLD with sleep disorders experienced muscle cramps [6]. Moreover, a decrease in muscle mass is likely to result in the onset of muscle cramps [31]. Thus, sleep disturbance may be associated with liver-related complications.

There was no difference in hepatic reserve between patients with normal sleep and those with sleep disturbance. However, many patients in the sleep disturbance group were receiving BCAAs, indicating that these patients actually had low hepatic reserve. Benzodiazepine is primarily metabolized in the liver, and the metabolic rate of the drug is slowed in patients with chronic liver disease [32]. As is known, patients with cirrhosis are typically sensitive to sleep medication, leading to HE following the administration of the drug. Further, in patients with CLD, sleep medications, such as gamma-aminobutyric acid agonists, are important precipitating factors for HE [33]. In our study, the number of patients receiving sleep medication was higher in the sleep disturbance group than in the normal sleep group. This may be because there are many causes, due to underlying disease, that cannot be released with sleep medication.

This study has some limitations. First, we only included inpatients, leading to the possibility of selection bias in the enrolled patients. Second, although this was a multicenter study, the number of patients was small. Third, as we did not evaluate the prevalence of covert HE, muscle cramps, or muscle mass, it was difficult to determine the cause of sleep disturbance. These factors should be investigated in future studies.

## 5. Conclusions

We revealed that the AIS, a simple questionnaire, can be used to assess sleep disturbance in patients with CLD. This is the first report to indicate that the AIS is an alternative questionnaire to the PSQI and that it can be a useful tool for detecting cirrhosis-related complications in patients with CLD.

## Figures and Tables

**Figure 1 medicina-58-00741-f001:**
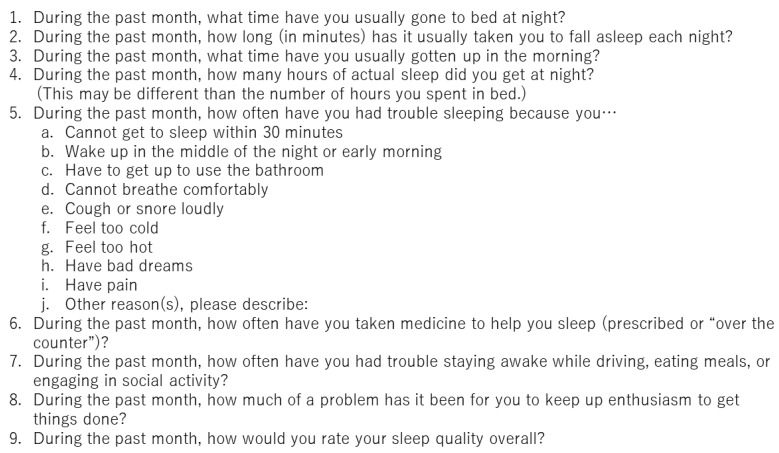
The contents of Pittsburgh sleep quality index.

**Figure 2 medicina-58-00741-f002:**
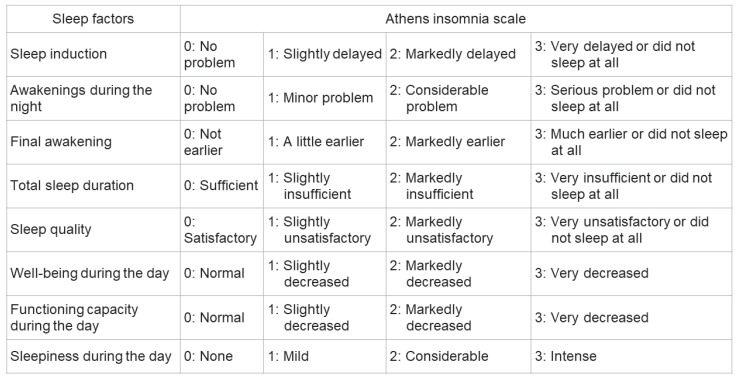
The contents of Athens insomnia score.

**Figure 3 medicina-58-00741-f003:**
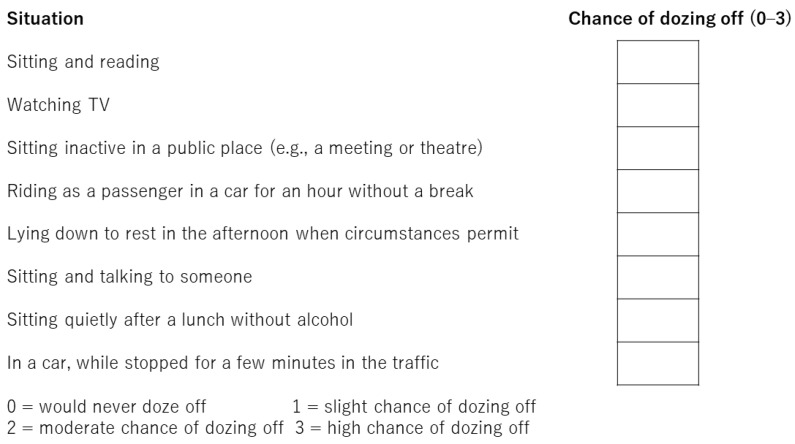
The contents of Epworth sleepiness scale.

**Figure 4 medicina-58-00741-f004:**
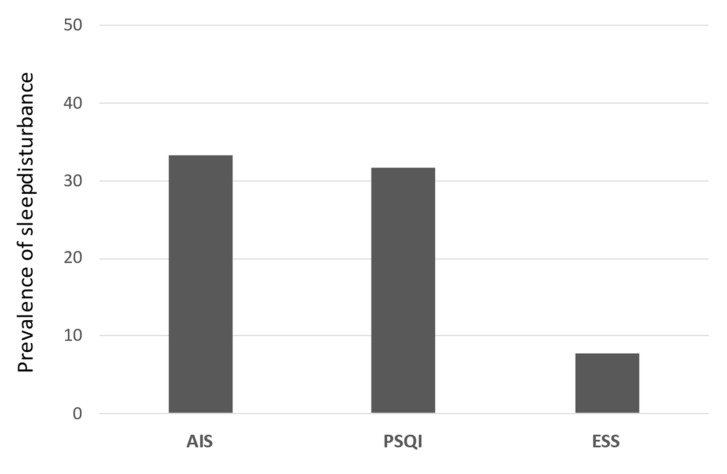
The incidence of sleep disturbance of each content. The proportion of patients with sleep disorder in AIS, PSQI, and ESS. AIS, Athens insomnia scale; PSQI, Pittsburgh sleep quality index; ESS, Epworth sleepiness scale.

**Figure 5 medicina-58-00741-f005:**
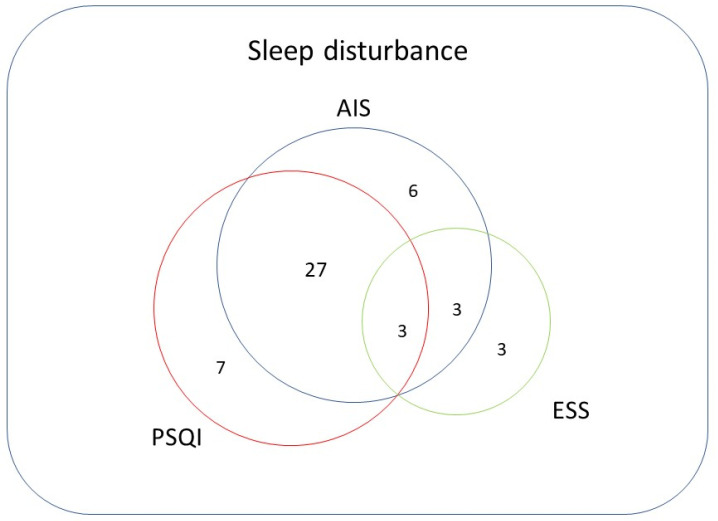
The Venn diagram of sleep disturbance, showing the relationships between AIS, PSQI, and ESS. AIS, Athens insomnia scale; PSQI, Pittsburgh sleep quality index; ESS, Epworth sleepiness scale.

**Table 1 medicina-58-00741-t001:** Baseline characteristics of PSQI scores (*n* = 117).

	Abnormal PSQI (*n* = 37)	Normal PSQI (*n* = 80)	*p*-Value
Age (years)	66.9 ± 9.5	70.1 ± 12.2	0.133
Sex (male:female)	24:13	54:26	0.834
Cause (B:C:NAFLD:ALD:AIH/PBC:others)	6: 3:11:11:4:2	9:18:22:19:6:6	0.477
Cirrhosis (yes:no)	36:1	68:12	0.060
HCC (yes:no)	22:15	51:29	0.685
Alb	3.6 ± 0.7	3.7 ± 0.6	0.422
T-Bil	1.6 ± 1.9	1.6 ± 3.6	0.876
ALBI score	−2.22 ± 0.71	−2.31 ± 0.52	0.461
mALBI grade (1:2a:2b:3)	12:8:16:2	26:19:29:4	0.967
Child–Pugh score (5:6:7:8:9:10 or more)	11:8:15:3:0:0	27:19:30:3:0:0	0.745
BUN	19.4 ± 11.4	17.4 ± 9.3	0.359
Cre	1.0 ± 0.6	0.9 ± 0.3	0.106
ALT	29.2 ± 19.7	57.4 ± 165.6	0.139
NH3	72.3 ± 51.9	51.5 ± 30.2	0.049 *
Hb	12.1 ± 1.5	12.6 ± 2.0	0.089
PT%	83.1 ± 21.1	81.2 ± 17.5	0.649
PLT	11.0 ± 6.6	12.5 ± 6.9	0.250
Sleep medication (yes:no)	14:23	7:73	<0.001 **
Shakuyakukanzoto (yes:no)	5:32	7:73	0.515
BCAA (yes:no)	19:18	24:56	0.039 *
Carnitine (yes:no)	3:34	0:80	0.030 *

PSQI, Pittsburgh sleep quality index; B, hepatitis B; C, hepatitis C; NAFLD, non-alcoholic fatty liver disease; ALD, alcoholic liver disease; AIH, autoimmune hepatitis; PBC, primary biliary cholangitis; HCC, hepatocellular carcinoma; Alb, albumin level; T-Bil, total bilirubin level; ALBI, albumin-bilirubin; mALBI, modified albumin-bilirubin; BUN, blood urine nitrogen; Cre, creatinine; ALT, alanine aminotransferase; NH_3_, ammonia; Hb, hemoglobin; PT% prothrombin time activity; PLT, platelet; BCAA, branched chain amino acid. * *p* < 0.05, ** *p* < 0.001.

**Table 2 medicina-58-00741-t002:** Baseline characteristics of AIS scores (*n* = 117).

	Abnormal AIS Score (*n* = 39)	Normal AIS Score (*n* = 78)	*p*-Value
Age (years)	66.4 ± 9.1	70.5 ± 12.3	0.044 *
Sex (male:female)	24:15	54:24	0.414
Cause (B:C:NAFLD:ALD:AIH/PBC:others)	4:2:13:13:5:2	11:19:20:17:5:6	0.082
Cirrhosis (yes:no)	38:1	66:12	0.058
HCC (yes:no)	21:18	52:26	0.225
Alb	3.7 ± 0.7	3.7 ± 0.6	0.912
T-Bil	1.5 ± 1.8	1.7 ± 3.7	0.783
ALBI score	−2.27 ± 0.67	−2.29 ± 0.54	0.919
mALBI grade (1:2a:2b:3)	12:8:16:2	26:19:29:4	0.967
Child–Pugh score (5:6:7:8:9:10 or more)	20:8:3:3:2:2	34:24:8:5:5:2	0.748
BUN	18.6 ± 11.0	17.7 ± 9.5	0.656
Cre	0.9 ± 0.4	0.9 ± 0.4	0.999
ALT	31.9 ± 20.3	56.8 ± 167.8	0.201
NH_3_	64.8 ± 37.5	55.0 ± 40.6	0.255
Hb	12.2 ± 1.4	12.6 ± 2.0	0.327
PT%	85.9 ± 20.3	79.7 ± 17.6	0.118
PLT	11.3 ± 7.9	12.4 ± 6.5	0.453
Sleep medication (yes:no)	11:28	10:68	0.071
Shakuyakukanzoto (yes:no)	9:30	3:75	0.002 **
BCAA (yes:no)	19:20	24:54	0.069
Carnitine (yes:no)	2:37	1:77	0.257

AIS, Athens insomnia scale; B, hepatitis B; C, hepatitis C; NAFLD, non-alcoholic fatty liver disease; ALD, alcoholic liver disease; AIH, autoimmune hepatitis; PBC, primary biliary cholangitis; HCC, hepatocellular carcinoma; Alb, albumin level; T-Bil, total bilirubin level; ALBI, albumin-bilirubin; mALBI, modified albumin-bilirubin; BUN, blood urine nitrogen; Cre, creatinine; ALT, alanine aminotransferase; NH_3_, ammonia; Hb, hemoglobin; PT% prothrombin time activity; PLT, platelet; BCAA, branched chain amino acid. * *p* < 0.05, ** *p* < 0.01.

**Table 3 medicina-58-00741-t003:** Baseline characteristics of combined AIS and PSQI scores.

	Abnormal AIS and PSQI (*n* = 30)	Normal AIS and PSQI (*n* = 71)	*p*-Value
Age (years)	66.1 ± 8.2	70.5 ± 12.2	0.041 *
Sex (male:female)	19:11	49:22	0.136
Cause (B:C:NAFLD:ALD:AIH/PBC:others)	3:2:9:11:4:1	8:18:18:17:5:5	0.024 *
Cirrhosis (yes:no)	30:0	60:11	0.468
HCC (yes:no)	18:12	48:23	0.681
Alb	3.6 ± 0.7	3.7 ± 0.6	0.893
T-Bil	1.6 ± 2.1	1.7 ± 3.8	0.703
ALBI score	−2.21 ± 0.71	−2.31 ± 0.52	0.931
mALBI grade (1:2a:2b:3)	9:7:12:2	24:18:26:3	0.729
Child–Pugh score (5:6:7:8:9:10 or more)	15:6:3:3:2:1	32:22:7:5:3:2	0.729
BUN	19.5 ± 12.3	17.6 ± 9.7	0.457
Cre	1.0 ± 0.5	0.9 ± 0.3	0.366
ALT	31.5 ± 21.2	60.4 ± 175.6	0.176
NH_3_	69.1 ± 40.7	51.3 ± 31.1	0.071
Hb	12.3 ± 1.5	12.7 ± 2.0	0.251
PT%	85.6 ± 20.4	80.4 ± 17.0	0.247
PLT	11.4 ± 7.1	12.7 ± 6.7	0.406
Sleep medication (yes:no)	11:19	7:64	0.003 **
Shakuyakukanzoto (yes:no)	5:25	3:68	0.048 *
BCAA (yes:no)	16:14	21:50	0.041 *
Carnitine (yes:no)	2:28	0:71	0.086

PSQI, Pittsburgh sleep quality index; AIS, Athens insomnia scale; B, hepatitis B; C, hepatitis C; NAFLD, non-alcoholic fatty liver disease; ALD, alcoholic liver disease; AIH, autoimmune hepatitis; PBC, primary biliary cholangitis; HCC, hepatocellular carcinoma; Alb, albumin level; T-Bil, total bilirubin level; ALBI, albumin-bilirubin; mALBI, modified albumin-bilirubin; BUN, blood urine nitrogen; Cre, creatinine; ALT, alanine aminotransferase; NH_3_, ammonia; Hb, hemoglobin; PT% prothrombin time activity; PLT, platelet; BCAA, branched chain amino acid. * *p* < 0.05, ** *p* < 0.01.

**Table 4 medicina-58-00741-t004:** Diagnostic ability of the AIS compared with the PSQI.

	Abnormal ESS Score	Normal ESS Score	Sum
Abnormal AIS score	6	33	39
Normal AIS score	3	75	78
Sum	9	108	117

AIS, Athens insomnia scale; ESS, Epworth sleepiness scale; PSQI, Pittsburgh sleep quality index.

**Table 5 medicina-58-00741-t005:** Comparisons between the ESS score and the PSQI or AIS score.

(A)			
	Abnormal AIS	Normal AIS	Sum
Abnormal PSQI	30	7	37
Normal PSQI	9	71	80
Sum	39	78	117
**(B)**			
	**Abnormal ESS Score**	**Normal ESS Score**	**Sum**
Abnormal PSQI score	3	34	37
Normal PSQI score	6	74	80
Sum	9	108	117

AIS, Athens insomnia scale; PSQI, Pittsburgh sleep quality index; ESS, Epworth sleepiness scale.

## Data Availability

Not applicable.

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
