# Peer review of "The Usefulness of the Athens Insomnia Scale for Evaluating Sleep Disturbance in Patients with Chronic Liver Disease Comparing with Pittsburgh Sleep Quality Index and Epworth Sleepiness Scale"

_medicina, 2022, doi:10.3390/medicina58060741_

Round 1

Reviewer 1 Report

Kawaratani et  al. aimed to determine whether the AIS and Epworth sleepiness scale (ESS) could be used as simple 21 alternative questionnaires for evaluating sleep disturbances in patients with CLD. A total of 117 patients with CLD were retrospectively evaluated. Patients with overt 23 hepatic encephalopathy were excluded. All patients were examined using the AIS, PSQI, and 24 ESS, and their responses to these questionnaires were statistically analyzed. The number 25 of patients diagnosed with sleep disturbance using the AIS, PSQI, and ESS were 39 (33.3%), 37 26 (31.6%), and 9 (7.7%), respectively.The authors concluded  that the AIS is an alternative 33 questionnaire to the PSQI and that it can be a useful tool for detecting cirrhosis-related compli-34 cations in patients with CLD.The  study  has  many  limitations. Some  issues  raised;

1-The  study  has  no  control  group. When comparing one test with another, it is necessary not only to compare it in the patient population, but also in a control group consisting of healthy volunteers. Thus, you will reach a clearer result.

2-Method section  is  too  short  and   is  not  clear. Method  section  must  give  more  information  about  tests (validation?).

 In sleep  studies, patients were classified into normal sleeping habit (PSQI score: 0–5 points), mild sleep disorder (6–8 points), moderate sleep disorder (9–11 points), and severe sleep disorder (12 points or greater) according to PSQI. The  authors reported  as a PSQI score ≥9 is indicative of sleep disturbance but  the  reference artciles  do not  support  this  information.

3- The discussion part is very superficial and does not cover the sleep medicine literature.

Thank you  for  giving ooportunity  to review  this  study.

Author Response

Reviewer 1

  1. The study has no control group. When comparing one test with another, it is necessary not only to compare it in the patient population, but also in a control group consisting of healthy volunteers. Thus, you will reach a clearer result.

→Thank you for your valuable comment. There had been a lot of report concerning of sleep disturbance comparing healthy and liver cirrhosis. Most of them shows that the prevalence of sleep disturbance is higher in liver cirrhosis. We added the sentence in the Introduction section. And it is difficult to get a healthy volunteer’s data in a short time. We’d like to get a control group’s data next time.

  1. Method section is too short and is not clear. Method section must give more information about tests (validation?).

→Thank you for your valuable comment. In accordance with the reviewer’s comment, we expanded the method section.

 In sleep studies, patients were classified into normal sleeping habit (PSQI score: 0–5 points), mild sleep disorder (6–8 points), moderate sleep disorder (9–11 points), and severe sleep disorder (12 points or greater) according to PSQI. The authors reported as a PSQI score ≥9 is indicative of sleep disturbance but the reference articles do not support this information.

→Thank you for your valuable comment. We changed the presentation of PSQI in accordance with the original range (original range (normal sleeping habit (PSQI score: 0–5 points), mild sleep disorder (6–8 points), moderate sleep disorder (9–11 points), and severe sleep disorder (12 points or greater)). As AIS scores of 3-5 are subclinical sleep disturbance, and which seems to correspond to PSQI 6-8, so, in this study, we defined PSQI ≥9, which assumes to correlate well with AIS ≥6, as sleep disturbance.

3- The discussion part is very superficial and does not cover the sleep medicine literature.

→Thank you for your valuable comment. We added the literatures about sleep medication in the discussion section, and enlarged the content in the revised manuscript.

Reviewer 2 Report

 The statistical difference can be expressed with symbols.

 Better one or two results in bar diagram 

Graphical abstract will improve the clarity of the content

The references should be cited as per journal format

Title can be modified with inclusion of ESS 

rationale for choosing the time period of the study

 Better include suggestions from Biostatistician  and its name as well in statistical analysis 

Author Response

Reviewer 2

The statistical difference can be expressed with symbols.

→Thank you for your comment. We added the symbols in the revised Tables (Table 1, 2, 3)

Better one or two results in bar diagram

→Thank you for your valuable comment. In accordance with the reviewer’s comment, we added the Figure 4 by using bar diagram.

Graphical abstract will improve the clarity of the content

→Thank you for your valuable comment. In accordance with the reviewer’s comment, we added Venn diagram for better understanding the relation between AIS, PSQI, and ESS (Figure 5).

The references should be cited as per journal format.

→Thank you for your valuable comment. We carefully changed the reference in accordance with the journal format.

Title can be modified with inclusion of ESS

→Thank you for your comment. We changed the title in accordance with the reviewer’s comment.

Rationale for choosing the time period of the study

→Thank you for your comment. We assumed to 40 cases per each facility (total 120 cases) for this study, and it took 6 months to collect the cases.

Better include suggestions from Biostatistician and its name as well in statistical analysis

→Thank you for your valuable comment. However, we have no budget to conduct the study and do not have the time and money to hire a Biostatistician to perform the additional analysis. We will schedule the analysis when we get a budget and do additional experiments in the future.

Round 2

Reviewer 1 Report

The responses  and corrections given do not meet the reviewer'comments clearly.